# Biomarker Correlations in PTSD: IL-18, IRE1, pERK, and ATF6 via Courtauld Emotional Control Scale (CECS)

**DOI:** 10.3390/ijms26157506

**Published:** 2025-08-03

**Authors:** Izabela Woźny-Rasała, Ewa Alicja Ogłodek

**Affiliations:** Collegium Medicum, Jan Dlugosz University in Częstochowa, Waszyngtona 4/8 Street, 42-200 Częstochowa, Poland

**Keywords:** activating transcription factor 6, inositol-requiring enzyme 1, Interleukin 18, phosphorylated extracellular signal-regulated kinase, post-traumatic stress disorder

## Abstract

Post-traumatic stress disorder (PTSD) is a chronic mental health condition resulting from exposure to traumatic events. It is associated with long-term neurobiological changes and disturbances in emotional regulation. Understanding the sociodemographic profiles, biomarkers, and emotional control in patients with PTSD helps to better comprehend the impact of the disorder on the body and its clinical course. An analysis of biomarkers such as Interleukin-18 (IL-18), Inositol-Requiring Enzyme 1 (IRE1), Phosphorylated Extracellular Signal-Regulated Kinase (pERK), and Activating Transcription Factor–6 (ATF-6) in PTSD patients with varying durations of illness (≤5 years and >5 years) and a control group without PTSD revealed significant differences. Patients with recently diagnosed PTSD (≤5 years) showed markedly elevated levels of inflammatory and cellular stress markers, indicating an intense neuroinflammatory response during the acute phase of the disorder. In the chronic PTSD group (>5 years), the levels of these biomarkers were lower than in the recently diagnosed group, but still significantly higher than in the control group. An opposite trend was observed regarding the suppression of negative emotions, as measured by the Courtauld Emotional Control Scale (CECS): individuals with chronic PTSD exhibited a significantly greater suppression of anger, depression, and anxiety than those with recent PTSD or healthy controls. Correlations between biomarkers were strongest in individuals with chronic PTSD, suggesting a persistent neuroinflammatory dysfunction. However, the relationships between biomarkers and emotional suppression varied depending on the stage of PTSD. These findings highlight the critical role of PTSD duration in shaping the neurobiological and emotional mechanisms of the disorder, which may have important implications for therapeutic strategies and patient monitoring.

## 1. Introduction

In the fifth edition of the Diagnostic and Statistical Manual of Mental Disorders (DSM-5), post-traumatic stress disorder (PTSD) is described as a disorder characterized by the presence of the following symptom clusters: intrusion symptoms (recurrent, involuntary, and intrusive distressing memories, dissociative reactions such as flashbacks, and physiological reactions to symbols or cues reminiscent of the trauma), avoidance symptoms (conscious avoidance of memories, thoughts, or feelings associated with the trauma), negative alterations in cognition and mood (including the inability to recall important aspects of the traumatic event due to severe stress), and alterations in arousal and reactivity [1,2]. To diagnose PTSD, these symptoms must persist for at least one month and cause clinically significant impairment in functioning. Emotional regulation is a key clinical feature of PTSD. In particular, suppression of emotional expression often represents a maladaptive strategy for coping with trauma. Although emotional suppression may provide short-term relief, as a recurring process, it can lead to the intensification of trauma symptoms and hinder the therapeutic process [3,4]. In this context, monitoring emotional regulation mechanisms is essential for a deeper understanding of pathological processes and for planning more effective therapeutic interventions [5]. A tool that enables the psychometric assessment of emotional inhibition is the Courtauld Emotional Control Scale (CECS), which allows for the quantitative determination of the tendency to suppress emotions in three key domains: anger, anxiety, and depression. The design of the CECS makes it possible to identify individual emotional regulation patterns that are relevant to both mental health and social functioning. However, it should be noted that the symptoms of anger, anxiety, and depression assessed using the CECS also have a neurobiological basis [6,7]. The scientific literature increasingly emphasizes that PTSD is not merely a disorder of emotional processing—it is also a complex pathophysiological state, associated with neuroinflammatory responses, the activation of cellular stress pathways, and disruptions in signaling within the central nervous system (CNS) [8,9,10,11]. The pathophysiology of PTSD focuses, among other factors, on the role of pro-inflammatory cytokines, which play a significant role in modulating CNS function and neuroinflammatory mechanisms [12,13]. Interleukin-18 (IL-18), a member of the IL-1 cytokine family, is particularly present in brain structures such as the amygdala and hippocampus—regions responsible for emotion regulation. Interleukin-18 (IL-18) acts as a potent inflammatory mediator through the activation of T-helper type 1 (Th1) lymphocytes, produced by activated microglial cells [14,15,16]. Importantly, this cytokine is capable of crossing the blood–brain barrier, which allows it to directly influence limbic structures crucial for emotion processing [17]. IL-18 activity is regulated by interaction with the IL-18 receptor complex, which consists of the interleukin-18 Receptor 1 (IL-18R1) subunit and the accessory protein Interleukin-18 Receptor Accessory Protein (IL-18RAP), responsible for initiating intracellular signaling [18,19]. IL-18 signaling is tightly controlled by the presence of the endogenous protein inhibitor Interleukin-18 Binding Protein (IL-18 BP), which limits excessive cytokine activation and helps maintain immune homeostasis in the CNS. Disruptions to this balance lead to an intensified inflammatory state observed in PTSD and other anxiety disorders [20].

Moreover, elevated IL-18 levels correlate with the severity of depression symptoms resulting from stress and with mechanisms responsible for the development of chronic pain [21]. Animal model studies have shown that chronic stress induces microglial activation in the hippocampus, leads to the activation of the NOD-like Receptor NLR (NLR) Family Pyrin Domain Containing-3 (NLRP3) inflammasome, and increases the expression of pro-inflammatory cytokines including Interleukin 1 beta (IL-1β) and IL-18 [22,23]. NLRP3 inflammasome activation proceeds in two stages. In the first—known as the priming signal—bacterial ligands recognized by Toll-like receptors (TLRs) initiate transcription of genes encoding pro-IL-1β, pro-IL-18, and NLRP3 through the activation of the nuclear factor NF-κB pathway. The second stage involves activation by Pathogen-Associated Molecular Patterns (PAMPs) and endogenous Danger-Associated Molecular Patterns (DAMPs), which trigger oligomerization of the inflammasome complex, including the NLRP3 protein, the adaptor protein ASC (apoptosis-associated speck-like protein containing a CARD), and pro-caspase-1 [24]. Caspase-1 activation leads to the transformation of inactive precursors of IL-1β and IL-18 into their active forms, which is a key element of the inflammatory neuroimmune response.

Another important mechanism triggered by trauma is endoplasmic reticulum (ER) stress, leading to the activation of the unfolded protein response (UPR). The main UPR sensor is inositol-requiring enzyme 1 (IRE1)—a kinase and endoribonuclease that, in response to the accumulation of misfolded proteins, initiates the splicing of the mRNA of the transcription factor X-box binding protein 1 (XBP1). The activated XBP1, in turn, regulates the expression of genes responsible for improving ER function, including chaperone proteins and enzymes that degrade defective proteins [25]. However, excessive or prolonged IRE1 activation may lead to the activation of pro-inflammatory signaling pathways such as c-Jun N-terminal kinase (JNK) and Nuclear Factor kappa-light-chain-enhancer of activated B cells (NF-κB), contributing to apoptosis and neuroinflammation [26]. In the context of PTSD, IRE1 activity gains particular importance—its ribonuclease function (RIDD) enables the selective degradation of double-stranded RNA and limits NLRP3 inflammasome activation. Inhibition of IRE1α function results in the accumulation of dsRNA, which may lead to excessive NLRP3 activation and cell death through pyroptosis [27]. Thus, IRE1α plays a crucial protective role by preventing excessive inflammatory response in neurons and microglia. The endoplasmic reticulum plays a fundamental role in protein synthesis, lipid metabolism, calcium storage, and redox balance. Its dysfunction—caused by oxidative stress, hypoxia, or mutations—can activate ER stress and initiate the three main UPR sensors: IRE1, protein kinase R-like endoplasmic reticulum kinase (pERK), and ATF6. Their function is to restore cellular equilibrium; however, prolonged UPR activation can lead to cell death. In PTSD, chronic ER stress and IRE1 overactivation disrupt neuron and microglial homeostasis, exacerbating neuroinflammatory and neurodegenerative processes. The IRE1 and UPR mechanisms are now viewed as promising therapeutic targets in neuropsychiatric disorders, including PTSD [28,29].

pERK plays a key role in the cellular response to ER stress, primarily by phosphorylating the translation initiation factor eIF2α (eukaryotic initiation factor 2 alpha). This phosphorylation temporarily halts general protein synthesis to reduce the burden of misfolded proteins in the ER. At the same time, pERK activation enables the selective translation of specific proteins such as activating transcription factor 4 (ATF4)—a transcription factor that regulates genes responsible for redox homeostasis, amino acid metabolism, and oxidative stress response [30,31]. ATF4, while initially protective, may—under conditions of prolonged stress—induce the expression of C/EBP homologous protein pro-apoptotic factor (CHOP), which acts as a pro-apoptotic factor. Excessive activation of the pERK–ATF4–CHOP pathway thus leads to neuronal apoptosis, impaired synaptic plasticity, and deteriorated cognitive and emotional functioning. Preclinical studies have shown that pERK overactivation and associated ER stress may play a significant role in PTSD pathogenesis, including increased hippocampal neuron vulnerability to damage, astrocyte dysfunction, and microglial activation [32].

The third key ER stress sensor is ATF6 (activating transcription factor 6)—a membrane-bound protein that, in response to ER stress, translocates from the ER to the Golgi apparatus [33]. There, it undergoes proteolysis by Site-1 protease (S1P) and Site-2 protease (S2P), leading to the release of its cytoplasmic domain, which acts as an active transcription factor. The activated form of ATF6 then translocates to the cell nucleus and regulates the expression of genes encoding chaperones, proteins involved in protein folding, and components of the endoplasmic reticulum-associated degradation (ERAD) system. Under physiological conditions, ATF6 activation constitutes part of the adaptive UPR; however, the prolonged activation of this pathway can lead to cellular homeostasis disturbances [34,35]. In the context of PTSD, preclinical studies suggest that ATF6 plays a dual role—on the one hand, it participates in neuron protection mechanisms, while on the other—under conditions of chronic stress—it may promote neurodegenerative processes, impaired neurogenesis, and dysregulation of the hypothalamic–pituitary–adrenal (HPA) axis. Furthermore, ATF6 influences the regulation of gene expression responsible for the inflammatory response, which is associated with microglial activation and enhanced neuroimmune response [36]. Microglia, as the primary component of the innate immune system of the CNS, serves neuroprotective functions under homeostatic conditions, but their chronic activation contributes to neurotoxicity, synaptic dysfunction, and disrupted neuronal communication. There is evidence that ATF6 dysfunction may lead to deregulated microglial function, which further exacerbates the emotional disturbances characteristic of PTSD [37].

The aim of this study is to identify relationships between the expression levels of selected molecular biomarkers—interleukin 18 (IL-18), IRE1 kinase, phosphorylated ERK (pERK), and transcription factor ATF6—and the results obtained on the individual subscales of the Courtauld Emotional Control Scale in patients diagnosed with post-traumatic stress disorder. By integrating psychometric data with biomolecular analyses, this study aims to deepen the understanding of neurobiological mechanisms responsible for deficits in emotional control and the persistent stress hyperreactivity characteristic of PTSD.

## 2. Results

### 2.1. Sociodemographic, Biomarker, and Emotional Control Profiles Across PTSD Status Groups

Understanding the sociodemographic, biomarker, and emotional control profiles of patients with PTSD provides critical insights into the disease’s impact and progression. Table 1 summarizes these characteristics across three groups—Past PTSD ≤ 5y (*N* = 33), Past PTSD > 5y (*N* = 31), and No PTSD controls (*N* = 28)—highlighting differences that may inform clinical management strategies.

#### 2.1.1. Impact of PTSD on Biomarker Levels and Emotional Control Profile in Patients

The presence of PTSD has a profound impact on both biomarker levels and emotional control, as reflected by the CECS scores, with clear differences emerging across our study groups.

When we examined biomarkers like IL-18, IRE1, pERK, and ATF6, we found that PTSD patients, whether recent (≤5y) or chronic (>5y), consistently showed elevated levels compared to controls without PTSD, with *p*-values from the Kruskal–Wallis test all below 0.001. Digging deeper with post hoc Dunn tests and Bonferroni correction (adjusted *p* < 0.0167), we saw that the Past PTSD ≤ 5y group had the highest IL-18 levels at a median of 125.2 ng/L (IQR 109.4–133.6), followed by the >5y group at 30.0 (30.2, IQR 20.8–73.4), and the No PTSD group at a much lower 9.0 (IQR 7.9–11.8). The post hoc analysis confirmed significant differences between all groups (a > b > c, adjusted *p* < 0.001), demonstrating that the inflammatory response, as measured by IL-18, is most intense in recent PTSD and, while it lessens over time, still remains higher than in controls.

A similar pattern held for IRE1, pERK, and ATF6, with the ≤5y group showing the highest levels—1608.0 pg/mL for IRE1 (IQR 1348.0–1987.0), 22.4 ng/L for pERK (IQR 20.2–24.2), and 24.0 ng/L for ATF6 (IQR 15.9–28.3)—compared to the >5y group (520.9, 7.5, and 7.2, respectively) and the No PTSD group (232.3, 3.5, and 3.1). Refer to Figure 1 for a visual representation of the results.

Turning to the CECS, which measures how much patients suppress negative emotions like anger, depression, and anxiety, we again saw striking differences. Higher scores mean greater suppression, and the Past PTSD > 5y group consistently showed the highest levels of suppression across all subscales.

For the Anger subscale, the >5y group had a median of 25.0 (IQR 22.0–27.0), compared to 16.0 (IQR 13.0–19.0) in the ≤5y group and just 7.0 (IQR 7.0–9.0) in the No PTSD group (*p* < 0.001, post hoc: a > b > c, adjusted *p* < 0.001).

The Depression subscale told a slightly different story: the >5y group scored 28.0 (IQR 26.0–28.0), significantly higher than both the ≤5y group at 10.0 (IQR 9.0–16.0) and the No PTSD group at 9.0 (IQR 7.0–14.0) (*p* < 0.001, post hoc: b < a, adjusted *p* < 0.017), with significant difference between the ≤5y and No PTSD groups (adjusted *p* = 0.020).

The Anxiety subscale mirrored the Anger pattern, with the >5y group at 27.0 (IQR 24.5–28.0), the ≤5y group at 16.0 (IQR 13.0–19.0), and the No PTSD group at 7.0 (IQR 7.00–9.00) (*p* < 0.001, post hoc: a > b > c, adjusted *p* < 0.017).

The total CECS score followed suit, with the >5y group at 79.0 (IQR 72.0–82.5), the ≤ 5y group at 42.00 (IQR 38.0–48.0), and the No PTSD group at 25.00 (IQR 22.8–29.0) (*p* < 0.001, post hoc: a > b > c, adjusted *p* < 0.001). These results demonstrate that PTSD patients, especially those with chronic PTSD, tend to suppress their negative emotions far more than controls, with the >5y group showing the most pronounced suppression across all domains. Refer to Figure 2 for a visual representation of the results.

#### 2.1.2. Influence of PTSD Duration on Biomarker Levels and Emotional Control Profile in Patients

The duration of PTSD significantly shapes the profile of both biomarkers and CECS scores, revealing distinct patterns between recent (≤5y) and chronic (>5y) PTSD patients. For the biomarkers, we observed a clear trend: the Past PTSD ≤ 5y group consistently had the highest levels of IL-18 (125.2 ng/L vs. 30.2 in >5y), IRE1 (1608.0 pg/mL vs. 520.9), pERK (22.4 ng/L vs. 7.5), and ATF6 (24.0 ng/L vs. 7.2), with post hoc tests confirming that these levels were significantly higher in the ≤5y group compared to the >5y group (adjusted *p* < 0.001 for all). This implies that the early years of PTSD are marked by intense inflammatory and stress responses, likely driven by the acute trauma’s impact on the body. Over time, these responses appear to attenuate, as seen in the lower levels in the >5y group, though they still remain elevated compared to the No PTSD group.

The CECS scores, however, tell a different story about emotional suppression. The Past PTSD > 5y group showed significantly higher suppression across all subscales compared to the ≤5y group. For the Anger subscale, the >5y group’s median of 25.0 was notably higher than the ≤5y group’s 16.00 (adjusted *p* < 0.001), and the same pattern held for the Anxiety subscale (27.0 vs. 16.0, adjusted *p* < 0.001). The Depression subscale showed the >5y group at 28.0, far exceeding the ≤5y group’s 10.0 (adjusted *p* < 0.001), with the total CECS score reflecting this trend (79.0 vs. 42.0, adjusted *p* < 0.001). These findings indicate that chronic PTSD patients suppress their negative emotions—anger, depression, and anxiety—much more than those with recent PTSD, indicating that over time, patients may develop stronger emotional suppression as a coping mechanism, which could contribute to the more severe dysthymic profile observed in the >5y group CECS total score (79.0 vs. 42.0).

### 2.2. Estimation of Correlations Between the Biomarker Levels and CECS Scores Across PTSD Status Groups

The correlations between biomarker levels (IL-18, IRE1, pERK, ATF6) and CECS scores (Anger, Depression, Anxiety subscales, and total score) reveal distinct patterns influenced by the presence and duration of PTSD, providing insights into the neurobiological and emotional dynamics of the disorder. For a comprehensive visualization of these correlations, refer to Figure 3, which presents correlation heatmaps for all parameters across the three PTSD status groups, highlighting the varying strengths and directions of associations.

In the No PTSD control group, correlations between biomarkers and CECS (not taking into account the correlations within biomarker parameters or within CECS scores) scores are generally weak, with the strongest positive association observed between ATF6 and CECS Depression (ρ = 0.425), inferring that higher ATF6 levels may be linked to increased suppression of depressive emotions in controls, though this correlation does not reach statistical significance after the Bonferroni adjustment (adjusted *p* = 0.006 for 8 comparisons per group). Biomarker inter-correlations in this group are moderate, such as between IRE1 and ATF6 (ρ = 0.595), indicating some shared pathways in stress response mechanisms, while correlations between biomarkers and CECS scores are mostly negligible (e.g., IL-18 with CECS Anger: ρ = −0.29).

In contrast, the Past PTSD > 5y group exhibits much stronger biomarker inter-correlations, with IL-18 showing high correlations with IRE1 (ρ = 0.879), pERK (ρ = 0.879), and ATF6 (ρ = 0.895), all significant (adjusted *p* = 0.006), reflecting a tightly coupled inflammatory and stress response in chronic PTSD. However, correlations between biomarkers and CECS scores in this group are weaker, with the strongest being IRE1 with CECS Anger (ρ = −0.31), indicating that chronic PTSD patients with higher IRE1 levels may suppress anger less, though this is not significant after adjustment.

The Past PTSD ≤ 5y group shows a similar pattern of strong biomarker inter-correlations (e.g., IL-18 with pERK: ρ = 0.771, adjusted *p* = 0.006), but the correlations with CECS scores are minimal, with the highest being IRE1 with CECS Anger (ρ = −0.25), indicating a slight tendency for higher IRE1 levels to be associated with less anger suppression. Comparing across groups using Fisher’s z-transformation, the correlation between IL-18 and IRE1 is significantly stronger in Past PTSD > 5y (ρ = 0.879) than in No PTSD (ρ = 0.409), with z = 2.87 (*p* = 0.004, two-tailed), demonstrating that PTSD enhances the coupling of inflammatory and stress pathways. Similarly, the correlation between CECS Anger and CECS Anxiety is notably stronger in Past PTSD ≤ 5y (ρ = 0.999) compared to No PTSD (ρ = −0.087), with z = 7.12 (*p* < 0.001), indicating that recent PTSD patients exhibit a near-perfect association between anger and anxiety suppression, unlike controls.

The duration of PTSD also influences these correlations; for instance, the correlation between IRE1 and CECS Anger shifts from ρ = −0.245 in Past PTSD ≤ 5y to ρ = −0.312 in Past PTSD > 5y (z = −0.31, *p* = 0.76), showing no significant change, but the overall pattern implies that chronic PTSD may slightly strengthen negative associations between biomarkers and emotional suppression.

Clinically, these findings stress that PTSD patients, particularly those with chronic exposure, exhibit a more synchronized biomarker response, potentially reflecting a sustained neuroinflammatory state, while emotional suppression patterns vary, with recent PTSD patients showing tighter links between anger and anxiety suppression, which may contribute to their emotional dysregulation. Additionally, the moderate correlation between ATF6 and CECS Depression in controls (ρ = 0.425) versus weaker associations in PTSD groups (e.g., ρ = 0.108 in >5y) infers that non-PTSD individuals may have a different emotional regulation mechanism influenced by stress response pathways, which could be explored for therapeutic targeting in PTSD management.

## 3. Discussion

Post-traumatic stress disorder constitutes a complex syndrome of symptoms arising from the dysregulation of the stress axis, the immune system, and neuronal plasticity mechanisms. Increasing evidence indicates that its underlying causes involve mutual interactions between chronic stress, inflammatory responses, and dysfunctions within the endoplasmic reticulum [38,39,40,41].

One of the key factors involved in this process is interleukin-18 (IL-18), a pro-inflammatory cytokine whose levels are elevated in individuals suffering from PTSD and correlate with the severity of anxiety symptoms, cognitive function deficits, and impaired emotional regulation [42,43]. In our study, we confirmed that IL-18 is significantly elevated in patients with PTSD and shows a positive correlation with the severity of the tendency to suppress emotional expression, as measured by the CECS. This observation aligns with previous studies indicating that IL-18 may influence the functioning of the hippocampus and amygdala through the activation of the NLRP3 inflammasome and the intensification of oxidative stress [44,45]. Importantly, IL-18 also affects neuroplasticity and neurogenesis pathways, disrupting learning and memory mechanisms—particularly under chronic stress conditions [46].

Another important aspect of our analyses was the assessment of the pERK (protein kinase RNA-like ER kinase) pathway, one of the main sensors of ER stress. Activation of pERK under conditions of protein-folding overload leads to translation inhibition and activation of the transcription factor CHOP, which may ultimately initiate neuronal apoptosis [47]. In our study, we found that pPERK levels significantly correlate with the severity of emotional suppression. These results are consistent with observations by Luhong L. et al., who demonstrated that pERK pathway activation in the hippocampus of rats exposed to chronic stress results in cognitive deficits and emotional disturbances [48].

At the same time, the increased expression of IRE1α—another crucial element of the ER stress response pathway—was observed. IRE1α regulates the splicing of XBP1 mRNA, mediating the cell’s adaptive response, but its excessive activation may lead to mRNA degradation and the activation of pro-apoptotic kinases such as JNK [49]. According to the literature, excessive IRE1α activity contributes to neuronal damage, particularly in limbic structures such as the hippocampus and amygdala [50]. In our study, IRE1α levels showed a significant correlation with the severity of emotional suppression, indicating its potential role in mechanisms of emotional avoidance in PTSD.

The third analyzed pathway—ATF6—plays a key role in regulating the expression of chaperone proteins involved in the cell’s adaptation to ER stress. However, under chronic overload of the UPR pathway, ATF6 activation may contribute to cognitive decline and neuronal degeneration [51]. In our study, ATF6 levels were shown to correlate with the intensity of emotional suppression tendencies, suggesting its involvement in regulating emotional expression in patients with PTSD. In light of previous findings, ATF6 overactivity may affect the functioning of prefrontal-limbic systems crucial for emotion control [52].

It is worth emphasizing that ER stress and inflammatory signaling mechanisms do not operate independently. On the contrary—ER stress and the activation of pro-inflammatory cytokines may mutually reinforce each other, creating a vicious cycle leading to neurodegeneration [53,54]. The synergistic action of IL-18 and activated UPR pathways leads to intensified apoptosis, mitochondrial dysfunction, and loss of neuronal homeostasis—which may translate into the observed clinical symptoms such as avoidance, irritability, and difficulties in emotional processing [55]. The integration of molecular data with psychological assessment, as in our case—analysis of the relationships between levels of IL-18, pPERK, IRE1α, and ATF6 and the severity of emotional suppression—constitutes a valuable contribution to the development of psychiatric biomarkers. This approach aligns with current trends in personalized medicine and translational psychiatry [56]. It not only enables the identification of potential therapeutic targets but also supports the development of individualized intervention strategies aimed at modulating ER stress and neuroinflammatory signaling pathways.

This study is not without limitations. The relatively modest sample size and the cross-sectional nature of the design inherently constrain the statistical power and limit the extrapolation of the findings to broader clinical populations with post-traumatic stress disorder. Furthermore, although the biomarker assays and psychometric tools employed were well-validated, the absence of longitudinal data precludes inferences regarding the temporal progression and causal directionality of the observed associations between endoplasmic reticulum stress markers, pro-inflammatory signaling, and emotional suppression.

## 4. Material and Methods

### 4.1. Characteristics of the Participants

To evaluate the impact of PTSD and its chronicity on clinical profiles, biomarker levels IL-18, IRE1, pERK, and ATF6, and CECS in male patients, comparing Past PTSD ≤ 5y (*N* = 33), Past PTSD > 5y (*N* = 31), and No PTSD controls (*N* = 28)

Secondary objective

To explore correlations between biomarkers and Courtauld Emotional Control Scale across groups, identifying neurobiological mechanisms and differences by PTSD status and duration.

### 4.2. Courtauld Emotional Control Scale

The study was conducted using the CECS developed by M. Watson and S. Greer, in the Polish adaptation by Z. Juczyński. The CECS is a questionnaire used to assess how individuals manage and express negative emotions, particularly anger, depression, and anxiety. It is a self-report measure consisting of 21 items across three subscales. Higher scores generally indicate a greater tendency to suppress emotions. The respondent rated the frequency of emotional experiences using a 4-point scale, ranging from “almost never” (1 point) to “almost always” (4 points). The scale has three subscales: one for anger, one for depression, and one for anxiety. Each subscale score is obtained by summing the scores of the items within that subscale. The total CECS score is the sum of the scores from all three subscales. The total score typically ranges from 21 to 84, with higher scores indicating a greater tendency to suppress emotions.

### 4.3. Blood Sampling and Serum Preparation

Peripheral venous blood samples were obtained from all participants at the initial assessment. The blood was drawn from the antecubital vein under standardized conditions using sterile vacutainer tubes without anticoagulants, allowing natural clot formation. After collection, the samples were left at room temperature for about 30 min to clot. They were then centrifuged at 2000× *g* for 10 min at 4 °C to separate the serum from the cellular components. The isolated serum was carefully transferred into sterile, nuclease-free cryogenic vials and immediately frozen at −80 °C to preserve cytokines and stress-related protein markers. All steps—including collection, processing, and storage—were conducted in accordance with strict biosafety and quality control standards to maintain sample consistency and integrity. Before analysis, serum samples were slowly thawed at 4 °C and gently mixed using a low-speed vortex to avoid protein damage. Each sample was diluted at a 1:3 ratio in the specific buffer recommended by the assay manufacturer to ensure accurate detection within the linear range. Samples were not subjected to repeated freeze–thaw cycles, and all assays were conducted in duplicate to enhance data reliability.

### 4.4. Biomarker Analysis Procedure

To measure the concentrations of IL-18, IRE1, pERK, and ATF6, enzyme-linked immunosorbent assays (ELISAs) were conducted using commercial kits that included calibration standards, blank controls, and serum samples diluted as recommended. The samples were added to microtiter plates pre-coated with specific capture antibodies for each target protein. The plates were incubated for 60 min at room temperature with continuous shaking at 300 rpm to promote the antigen–antibody interaction. After incubation, unbound substances were washed away, and biotin-labeled detection antibodies were applied. A second 60 min incubation under identical conditions followed. The plates were then washed again, and streptavidin conjugated with horseradish peroxidase (HRP) was added and incubated for 30 min to enable signal generation. After a final wash step, 100 µL of substrate solution was introduced into each well. The enzymatic reaction proceeded for 10 min at room temperature and was stopped with the addition of an acidic stop solution. Absorbance was measured at 450 nm using a microplate reader to determine analyte concentrations.

### 4.5. Assay Sensitivity and Standardization

For each biomarker, calibration curves were constructed using recombinant protein standards. Details regarding reagent catalog numbers, detection limits, and sensitivity parameters are provided below.

IL-18: Assay range 0.6–100 ng/L; sensitivity 0.537 ng/L; (Shanghai, China); Catalog No 201-12-0148

IRE1: Assay range 8–2000 pg/mL; sensitivity 7.362 pg/mL; (Shanghai, China); Catalogue No 201-12-6275

pERK: Assay range 0.08–20 ng/mL; sensitivity 0.073 ng/mL; (Shanghai, China); Catalogue No 201-12-0720

ATF6: Assay range 15–3600 ng/L; sensitivity 11.625 ng/L; (Shanghai, China); Catalogue No 201-12-3083

### 4.6. Statistical Analysis

Statistical analyses were conducted using R (version 4.3.1). Descriptive statistics were reported as medians with interquartile ranges (IQR) for all continuous variables, including age, employment duration, biomarker levels, and CECS scores, due to their non-normal distributions as confirmed by the Shapiro–Wilk test (*p* < 0.05 for all variables). Group comparisons for these variables were performed using the Kruskal–Wallis rank sum test, with *p* < 0.05 indicating significant differences across the three groups (Past PTSD ≤ 5y, Past PTSD > 5y, No PTSD). Post hoc pairwise comparisons were conducted using the Dunn test with Bonferroni correction to control for multiple comparisons (adjusted *p* < 0.0167 for 3 pairwise comparisons per variable). The compact letter display (CLD) notation was used to summarize significant differences, where groups with different superscript letters (a, b, c) differ significantly, and those sharing the same letter do not. Effect sizes for significant Kruskal–Wallis tests were calculated using the rank-biserial correlation (r), interpreted as small (0.1), medium (0.3), or large (0.5).

Correlations between biomarker levels and CECS scores were assessed using Spearman’s rank correlation coefficient (ρ), given the non-normal distribution of the data and the ordinal nature of CECS scores. Correlations were calculated separately for each group, resulting in 64 correlations per group (8 parameters × 8 parameters). Significance was determined using unadjusted *p*-values, with a Bonferroni-adjusted significance threshold of *p* < 0.000781 (0.05/64) to account for multiple comparisons within each group. For significant correlations, 95% confidence intervals (CI) were computed using the Fisher z-transformation method to provide a measure of precision. Comparisons of correlation coefficients between groups were performed using Fisher’s z-transformation, with two-tailed p-values reported to assess whether the strength of correlations differed significantly across groups.

Statistical significance for all tests was set at *p* < 0.05. All *p*-values are two-tailed, and results are presented with a focus on clinical relevance, including effect sizes and confidence intervals where appropriate.

### 4.7. Data Collection

Sociodemographic data, including age and duration of employment in hazardous conditions, were collected for all participants. Biomarker levels, specifically IL-18 (ng/L), IRE1 (pg/mL), pERK (ng/L), and ATF6 (ng/L), were measured to assess inflammatory and stress response pathways. Emotional control was evaluated using the Courtauld Emotional Control Scale (CECS), which measures the suppression of negative emotions across three subscales—Anger, Depression, and Anxiety—along with a total score. Higher CECS scores indicate greater suppression of emotions. All data were collected in accordance with ethical guidelines, and informed consent was obtained from all participants.

## 5. Conclusions

The results of the conducted study provide significant evidence for the existence of associations between endoplasmic reticulum stress biomarkers (pPERK, IRE1α, ATF6) and the pro-inflammatory cytokine IL-18 with emotional regulation strategies in patients with post-traumatic stress disorder. The observed correlation between the elevated expression of these proteins and a heightened tendency to suppress emotions suggests that cellular stress response mechanisms and inflammatory signaling may jointly influence the neurobiological basis of avoidant behavior and emotional dysregulation characteristic of PTSD.

Our findings confirm that combining molecular analysis with precise clinical assessment can reveal new aspects of PTSD pathophysiology and contribute to identifying therapeutic targets, including the modulation of UPR pathways and neutralization of pro-inflammatory cytokines. The role of the pERK, IRE1α, and ATF6 pathways in maintaining neuronal homeostasis and their interaction with the immune system opens the possibility for new neuroprotective interventions, such as selective ER stress kinase inhibitors or immunomodulators targeting IL-18 and the NLRP3 inflammasome.

In light of these findings, future research should focus on the following: long-term monitoring of biomarker level changes in response to treatment, evaluating their utility as indicators of pharmacological and psychotherapeutic intervention effectiveness, and validating these results in larger, more diverse clinical populations. Understanding the role of complex interactions between cellular stress, inflammation, and emotion regulation may contribute to a more precise biological classification of PTSD and support the development of modern, individualized therapeutic strategies based on molecular biomarkers.

## Figures and Tables

**Figure 1 ijms-26-07506-f001:**
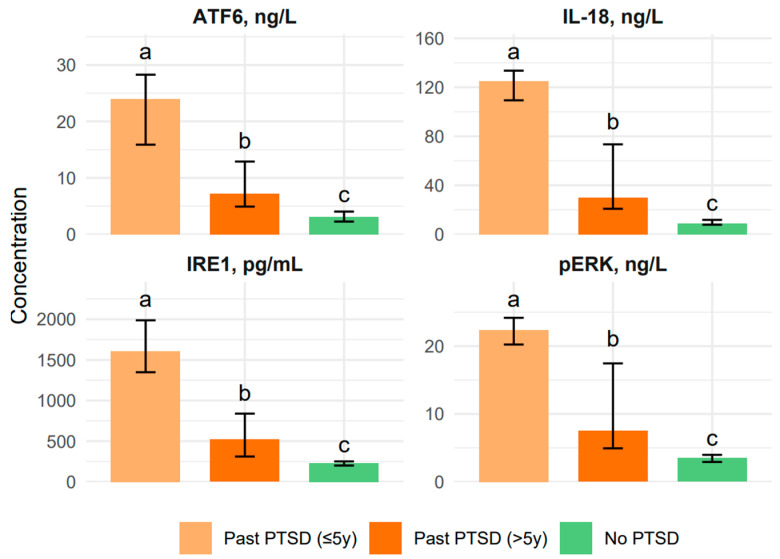
Biomarker Distributions Across PTSD Status Groups.

**Figure 2 ijms-26-07506-f002:**
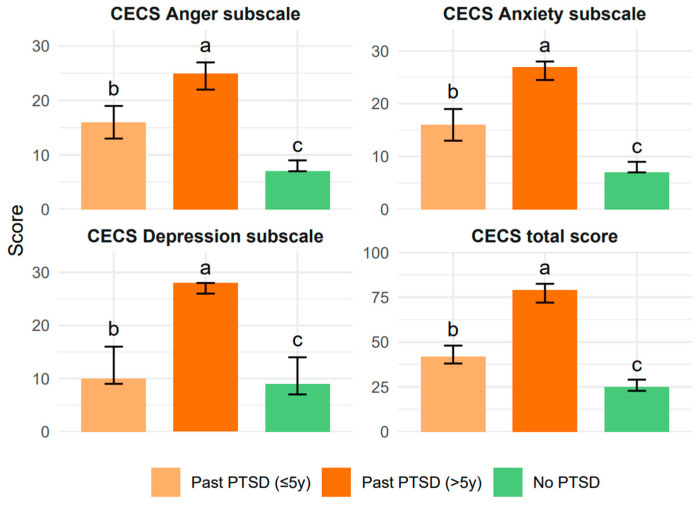
CECS parameter distributions across PTSD status groups.

**Figure 3 ijms-26-07506-f003:**
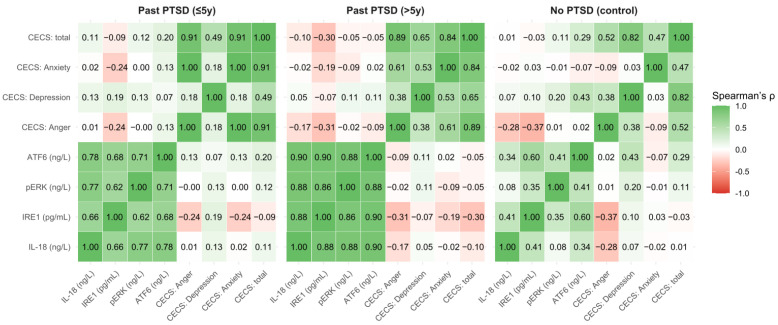
Correlation heatmaps of biomarkers and CECS parameters across PTSD status groups.

**Table 1 ijms-26-07506-t001:** Demographic, biomarker, and questionnaire (CECS) characteristics by PTSD status.

** *Characteristic* **	** *Total* ** ** *(N = 92)* **	** *Past PTSD (≤5y) (N = 33)* **	** *Past PTSD (>5y)* ** ** *(N = 31)* **	** *No PTSD (Control)* ** ** *(N = 28)* **	***p*-Value**	** *Post hoc* **
**Demographic characteristics, median (IQR)**						
Age, years	34.0 (28.8–41.0)	34.0 (31.0–41.0)	36.0 (29.5–41.0)	33.5 (24.3–41.5)	0.524	-
Employment in hazardous conditions, years	10.0 (6.0–14.25)	11.0 (7.0–14.0)	10.0 (7.5–15.0)	10.0 (3.0–14.0)	0.418	-
**Biomarker Levels, median (IQR)**						
IL-18, ng/L	36.2 (11.1–112.1)	125.2 (109.4–133.6) ^a^	30.15 (20.8–73.4) ^b^	9.0 (7.9–11.8) ^c^	**<0.001**	a > b > c
IRE1, pg/mL	577.3 (246.8–1379.0)	1608.0 (1348.0–1987.0) ^a^	520.9 (310.7–838.9) ^b^	232.3 (199.9–251.0) ^c^	**<0.001**	a > b > c
pERK, ng/L	10.0 (3.8–21.7)	22.4 (20.2–24.2) ^a^	7.5 (4.9–17.5) ^b^	3.5 (2.9–4.0) ^c^	**<0.001**	a > b > c
ATF6, ng/L	9.5 (3.6–17.5)	24.0 (15.9–28.3) ^a^	7.2 (4.9–12.9) ^b^	3.05 (2.3–4.0) ^c^	**<0.001**	a > b > c
**Questionnaire (c) parameters, median (IQR)**						
CECS Anger subscale	16.5 (9.0–22.3)	16.0 (13.0–19.0) ^b^	25.0 (22.0–27.0) ^a^	7.0 (7.0–9.0) ^c^	**<0.001**	a > b > c
CECS Depression subscale	14.0 (9.0–26.0)	10.0 (9.0–16.0) ^b^	28.0 (26.0–28.0) ^a^	9.0 (7.0–14.0) ^c^	**<0.001**	a > b > c
CECS Anxiety subscale	16.50 (10.9–25.0)	16.0 (13.0–19.0) ^b^	27.0 (24.5–28.0) ^a^	7.0 (7.0–9.0) ^c^	**<0.001**	a > b > c
CECS total score	43.0 (30.0–72.0)	42.0 (38.0–48.0) ^b^	79.0 (72.0–82.5) ^a^	25.0 (22.8–29.0) ^c^	**<0.001**	a > b > c

**Notes**: All variables are reported as median (IQR) due to non-normal distributions (Shapiro–Wilk test, *p* < 0.05). *p*-values were calculated using the Kruskal–Wallis rank sum test; *p* < 0.05 indicates significant differences across groups. Post hoc pairwise comparisons were performed using the Dunn test with Bonferroni correction (adjusted *p* < 0.0167 for 3 comparisons); groups with different superscript letters (a, b, c) differ significantly (adjusted *p* < 0.0167), while those sharing the same letter do not. The Post hoc column summarizes the direction of significant differences (e.g., a > b > c indicates Past PTSD ≤ 5y > Past PTSD > 5y > No PTSD). CECS = Courtauld Emotional Control Scale–CECS; IQR = interquartile range; PTSD = post-traumatic stress disorder.

## Data Availability

All data and analysis are available within the manuscript, or upon request to the corresponding author.

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
