# Peer review of "Biomarker Correlations in PTSD: IL-18, IRE1, pERK, and ATF6 via Courtauld Emotional Control Scale (CECS)"

_ijms, 2025, doi:10.3390/ijms26157506_

Round 1
Reviewer 1 Report
Comments and Suggestions for Authors
This is an excellent examination of a number of biomarkers in PTSD. Inflammatory interleukins (in particular IL-18) prove to be elevated particularly among those with PTSD duration below 5 years. It is a good point that the longterm relationships are different than they are in the more acute phases although of course 5 years is a long period (which could be discussed). The group sizes do not allow a more fine-grained division into duration groups. Accordingly it makes sense to divide the patient population according to duration but there could be more discussion regarding the interpretation. Many patients with PTSD improve (and "disappear" from the cohort) which means that a series of cross-sectional correlations is not comparable to a longitudinal study. This should also be discussed.
Suppression of emotion is an interesting variable that the authors have focused on. This is a kind of variable which has been on stage for a very long time although names have varied. Denial has been a more common label in the past. That should perhaps also be discussed a little.
The number of biomarkers of inflammation is increasing all the time and my own knowledge about some of the more newly established molecules is vague.
The authors´ handling of experimental procedures, statistics and language is excellent.
Author Response
Dear Reviewer,
We sincerely thank you for your valuable comments and positive evaluation of our manuscript.
We agree that discussing the study limitations is essential. Therefore, we have added a section in the Discussion addressing the cross-sectional design of our study and its related interpretative constraints.
Regarding the topic of emotional suppression, we appreciate your insightful remarks on the historical and conceptual aspects of this phenomenon. Due to the complexity and importance of this subject, we plan to dedicate a separate, more detailed article to it. In the current manuscript, we aimed only to highlight emotional suppression as a significant mechanism of emotional regulation in PTSD and therefore did not expand this topic extensively to maintain clarity and focus.
Once again, thank you for your constructive suggestions, which have helped us improve the scientific quality and clarity of our paper.
With kind regards,
The Autors
Reviewer 2 Report
Comments and Suggestions for Authors
This manuscript presents an important investigation into the biological and emotional correlates of PTSD, specifically examining the roles of IL-18, IRE1, pERK, and ATF6 alongside emotional suppression measured by the Courtauld Emotional Control Scale (CECS). By stratifying participants into acute (≤5 years) and chronic (>5 years) PTSD groups, the authors reveal differences in neuroinflammatory profiles and emotional regulation patterns across the disease course. The strongest biomarker correlations were observed in chronic PTSD, while emotional suppression increased with illness duration. I think a minor revision before publication.
Two small part:
1. Address limitations briefly in the discussion:
-
For example, note that cross-sectional design limits causal interpretation.
-
Suggest the potential influence of medication or comorbidities (if applicable)
2. editorial polish:
-
There are some grammatical inconsistencies in the abstract (e.g., “triggered by exposure” might read more clearly as “resulting from exposure”).
-
Consider rewording “much stronger suppression” to “significantly greater suppression” for more academic tone.
Author Response
Dear Reviewer,
We sincerely thank you for your valuable comments and positive evaluation of our manuscript.
We agree that discussing the study limitations is essential. Therefore, we have added a section in the Discussion addressing the cross-sectional design of our study and its related interpretative constraints. The grammatical improvements you suggested have also been implemented. Please note that our study did not involve any investigation of medication use among the patients.
Once again, we appreciate your constructive feedback, which helped improve the clarity and quality of our work.
With kind regards,
The Authors